# Effects of Human Harvesting, Residences, and Forage Abundance on Deer Spatial Distribution

**DOI:** 10.3390/ani14131924

**Published:** 2024-06-29

**Authors:** Hayato Takada, Keita Nakamura

**Affiliations:** 1Mount Fuji Research Institute, Yamanashi Prefecture Government, 5597-1 Kenmarubi, Kamiyoshida, Fujiyoshida 403-0005, Yamanashi, Japan; k.nakamura@mfri.pref.yamanashi.jp; 2Wildlife Management Center, Tokyo University of Agriculture and Technology, 3-5-8 Saiwaicho, Fuchu 183-8509, Tokyo, Japan

**Keywords:** habitat selection, landscape of fear, harvesting, distribution, sika deer, *Cervus nippon*, forage availability, foraging pressure, ungulates, large herbivore

## Abstract

**Simple Summary:**

Investigating the spatial distribution of wildlife contributes to understanding the adaptation of wildlife and conservation management. We assessed the summer spatial distribution of sika deer *Cervus nippon* in relation to human harvesting and other factors, such as human residences, forage abundance, and cover. Spatial distribution of deer was significantly biased to non-harvesting areas and far from residential areas, suggesting that the deer avoid riskier spaces by establishing a landscape of fear. High-quality food resources (deciduous broad-leaved trees and forbs) were more abundant in harvesting areas than in non-harvesting areas, suggesting that foraging pressure by deer reduce them. However, deer frequently used higher dwarf bamboo abundance areas, suggesting that the dwarf bamboo is an alternative food resource in non-harvesting areas. Our results suggest that human activities shifted the spatial distribution of deer to areas far from human activities, such as subalpine/alpine zones, which may increase damage to vulnerable ecosystems due to severe foraging pressure.

**Abstract:**

It has been known that harvesting by humans strongly influences individual within-home range habitat selection of many deer species; however, little is known about the effect of harvesting on coarse-scale habitat selection (i.e., spatial distribution). We examined the summer spatial distribution of sika deer *Cervus nippon* in relation to human harvesting and other factors, such as human residences, forage abundance, and cover, using pellet group counts at Mount Fuji, central Japan, in 2018. In the study area, harvesting is conducted at medium elevation areas throughout the year, but not at high or low elevation areas where access is difficult or harvesting is prohibited. Spatial distribution of deer was significantly biased to non-harvesting areas and far from residential areas, suggesting that they avoid riskier spaces by establishing a landscape of fear. High-quality food resources (deciduous broad-leaved trees and forbs) were more abundant in harvesting areas than in non-harvesting areas, suggesting that foraging pressure by deer reduce them. However, there were no differences in abundances of more fibrous dwarf bamboo between harvesting and non-harvesting areas, and spatial distribution of deer was significantly biased to higher dwarf bamboo abundance areas, suggesting that the dwarf bamboo is an alternative food resource in non-harvesting areas where supplies of high-quality food were limited. Our results suggest that human harvesting pressure and residences shifted the spatial distribution of deer from the montane forests to subalpine/alpine zones, which may increase damage to vulnerable ecosystems due to severe foraging pressure.

## 1. Introduction

Predation risk is one of the most lethal selection pressures and strongly influences the habitat selection of prey species, such as those of the deer family (Cervidae) [1,2,3,4,5]. Hunting by predators is not guaranteed to be successful, so surviving prey perceive risky domains (e.g., spaces or times with a high likelihood of predation) by establishing a “landscape of fear” [6,7]. Prey exposed to high predation risk prioritize predation avoidance over other factors, such as food resources or thermal conditions, and thus spatially avoid risky habitats [8,9]. In addition, the acquisition of high-quality resources and predation risk avoidance are often in a trade-off relationship, with safer habitats tending to have poorer food resources and riskier habitats tending to have richer food resources [10,11,12]. For many deer species, harvesting by humans is a major source of mortality and strongly influences various behavioral traits, including habitat selection, and generates a landscape of fear [13,14,15]. For example, individual roe deer *Capreolus capreolus* avoided riskier habitats of croplands during the daytime and in the hunting seasons, when the probability of lethal encounters is higher [16]. Similarly, at the start of the hunting season, male red deer *Cervus elaphus* immediately switched to habitat with more concealing cover, which provided security [17]. However, most of these studies of deer focused on habitat selection at an individual home range scale, and the effect of harvesting on coarse-scale habitat selection (i.e., spatial distribution) is still unclear.

From the 1990s to the present, sika deer *Cervus nippon* populations have been erupting and expanding their distribution in Japan [18]. Overpopulation of sika deer, through their foraging pressures, eliminates understory vegetation, prevents forest regeneration, and causes soil erosion in many Japanese forested areas [19,20]. In the last two decades, sika deer have been expanding their range into subalpine and alpine zones, where their distribution had not been confirmed, and have been exerting an effect on subalpine and alpine ecosystems [21,22,23,24]. To mitigate the damage caused by sika deer, from 2000 to the present, between 100,000 and 700,000 sika deer have been harvested each year in Japan, but the damage caused by this species continues [25]. Understanding the factors that determine habitat selection of sika deer is essential for biodiversity conservation and their population management.

There are a number of studies on habitat use of sika deer in relation to acquisition of food resources, cover from climatic conditions, or avoidance of human activities at some spatial scales, such as a landscape scale [26,27] and home range scale [28,29]. For example, this species prefers habitats providing richer food resources, such as clearings [30,31,32], grasslands or meadows [33,34], forest edges [35,36,37], pastures [28,38], croplands [29], and dwarf bamboo communities [39]. This species prefers areas with little snow cover and evergreen coniferous forests that provide cover from adverse climatic conditions in winter [29,40], and avoids areas near human activity, such as human residential areas and roads [27,37,41]

There are no wild predators of adult sika deer in Japan (wolves, *Canis lupus*, have been extinct for over 100 years [42]), but Asiatic black bears occasionally prey on fawn [43]. Thus, harvesting by humans is the main source of mortality for this species, and is likely to exert a considerable selection pressure. Kamei et al. [28] investigated within-home range scale habitat selection of adult females using GPS collar tracking in harvesting areas, showing that the sika deer avoided riskier open pastures during harvesting periods. Although harvesting by humans may also influence coarse-scale habitat selection, no study has demonstrated the effect of human harvesting to date. Medium elevation areas of montane zones are where sika deer populations first increased, and there are many forest roads that are easily accessed for humans, and thus, harvesting by humans tends to be carried out constantly in such areas. Conversely, high elevation areas of subalpine/alpine zones with few forest roads are difficult to access for humans, and human harvesting is prohibited at low elevation areas near human residential areas. Therefore, sika deer can learn which spaces are riskier, and their spatial distribution may shift to safer areas without harvesting, such as high mountains or near human residential areas, by establishing a landscape of fear.

As mentioned above, high deer densities lead to poor understory vegetation due to foraging pressure, and, in particular, broad-leaved trees and forbs that are intolerant to foraging pressure tend to be more affected than graminoids that are tolerant [19,44,45]. Moreover, leaves of deciduous broad-leaved trees and forbs tend to be more digestible and contain more protein than graminoids [46,47], and are the preferred food items for sika deer [48]. Therefore, in riskier areas (i.e., harvesting areas), they may have little effect on vegetation, whereas in safer areas (i.e., non-harvesting areas), foraging pressure may have severe effects on broad-leaved trees and forbs rather than graminoids.

The aim of this study was to reveal the relationship between summer spatial distribution of sika deer and human harvesting and other factors, such as forage abundance, cover, and human residential areas, in Mount Fuji, including from low to high elevation areas, in central Japan. We hypothesized that human harvesting shifted the spatial distribution of sika deer from the montane forests to subalpine/alpine zones or near urban areas, which cause severe effects on vegetation in non-harvesting areas through foraging pressure. To test this hypothesis, we evaluate the following two predictions: (P1) they prioritize predation avoidance over other factors, that is, their spatial distribution is biased towards non-harvesting areas (i.e., high or low elevation areas) to avoid human harvesting in mid-elevation areas, and (P2) non-harvesting areas have poorer food resources, especially in broad-leaved trees and forbs, than harvesting areas due to severe foraging pressure.

## 2. Materials and Methods

### 2.1. Study Area

The study was carried out on the north-facing slope of Mount Fuji, Yamanashi Prefecture, central Japan; the study area covered approximately 131 km^2^ at an elevation of 880–2250 m (Figure 1). The climate of this area includes cool temperate zones such as the montane and subalpine zones. The overstory mainly consisted of coniferous trees such as Veitch’s silver-fir *Abies veitchii* Lindl. (32.1%), Japanese larch *Larix kaempferi* Lamb. (29.3%), Japanese red pine *Pinus densiflora* Siebold and Zucc. (22.4%), and other coniferous trees (6.8%), including Japanese cypress *Chamaecyparis obtusa* Siebold and Zucc. and Japanese cedar *Cryptomeria japonica* Thunb. Forests were split into three types: Veitch’s silver-fir, Japanese larch, and Japanese red pine. We did not survey other coniferous forests, grasslands, or volcanic deserts, which were present only in small patches. The few human residential areas were sparsely distributed, mainly at lower elevations (<1100 m), and there were urban areas outside the survey area at even lower elevations (Figure 1).

During the 1960s and 1980s, the population density of sika deer in this area seemed to be relatively low, and their range seemed to be limited to the montane zones (i.e., mid-elevation areas, [49,50]). Since the 1990s, their range has expanded and their population size has increased, and now they inhabit a wide area from near urban areas to alpine zones [51,52]. The relative population density of sika deer in this area doubled in the decade from 2009 to 2019 [53]. Effects on vegetation or other animals by sika deer have been confirmed in subalpine coniferous forests [23,24] and alpine zones [54]. Major food items for sika deer in this area during summer are leaves of dwarf bamboos, deciduous broad-leaved trees, or forbs [54]. Their home range size during summer ranged from 50 to 100 ha, estimated by 100% minimum convex polygon methods using GPS collar tracking data [55]. Most of the sika deer living in this area perceive humans as a potential danger, showing vigilance, flight, and various antipredator behaviors in the presence of humans [55].

In the study area, for the purpose of sika deer population control, government-led harvesting has been conducted throughout the year from 2005 to the present (2023), and peak season of sika deer harvesting is from May to October without snow cover. The government (Yamanshi Prefecture) outsources the harvesting to the hunting associations of Yamanashi Prefecture (“Ryouyukai”). Both male and female sika deer are targeted for harvesting, and the sex ratio of harvested deer is approximately one to one. Since the study area is located in a game reserve, general hunters are prohibited from hunting. Harvesting is often conducted by teams of approximately 10 people using guns (termed “maki-gari”) during the daytime, which is a traditional hunting method in Japan, four times a month. Maki-gari is a hunting method in which several hunters drive sika deer herds, while others ambush and shoot the sika deer on their escape route. A total of 4657 sika deer were harvested from 2015 to 2021 (mean of 665 individuals per year). We conducted interviews with Ryouyukai that conduct harvesting in this area, and mapped areas that were harvested (harvesting area) and those that were not (non-harvesting area). Harvesting was mainly carried out at medium elevation areas (approximately 1100–1500 m) with many forest roads, and was not carried out in high elevation areas without forest roads or in low elevation areas near residential areas where hunting is prohibited (Figure 1).

### 2.2. Deer Spatial Distribution

To evaluate the spatial distributions of sika deer, we counted their fecal pellet groups between June and August (including peak season of sika deer harvesting) 2018. Sixty belt transects were set with lengths of 500–800 m (total length, 33,323 m; mean length, 555.4 m) and widths of 1 m (Figure 1). The nearest distance between each transect was at least 500 m. The study area was divided into 1 km grids (similar size as sika deer home ranges), and the grid ID in which each transect was located was recorded. Understory and/or other environmental conditions differ depending on forest type, which may influence sika deer defecation rate. Therefore, to minimize effect of defecation rate differences according to habitat, we set each transect to include only one forest type. To minimize differences in detection rate of sika deer pellet groups among transects, we slowly walked transects (for 50 to 90 min/transect), especially transects with dense understory, during daytime, and set transects with narrow width (1 m). To minimize differences in decay rate of pellet group among transects, we only counted fresh pellet groups, which were excreted by sika deer within a few days. Fresh pellets were defined by the surface being shiny and covered in a layer of slime. Sika deer fecal pellets were distinguishable from Japanese serow *Capricornis crispus* pellets because sika deer fecal pellets are dispersed whereas serow fecal pellets form dung piles [56].

Pellet counts have been widely used to express the habitat use of ungulates [57,58,59]. In general, sika deer defecate randomly without a latrine, and the longer they stay in a place, the more likely they are to defecate [39]. This means that the pellet counts may be suitable for evaluating the habitat use of sika deer like some other cervids [60,61,62]. Thus, we considered the number of pellet groups to represent sika deer relative habitat use. Sika deer pellet group density (pellet group number/100 m) was used as a proxy for sika deer habitat use in each transect

### 2.3. Environmental Variables

To evaluate environmental characteristics for each transect, we recorded several environmental variables that may influence habitat use of sika deer at 100 m intervals for each transect (i.e., the number of sampling points in each transect ranged from 6 to 8 depending on the length of the transect): distance to nearest human residential area (m), biomass index (hereafter “BI”) of forage for sika deer, and visibility score. Transects that even slightly overlapped with harvesting areas were defined as “harvesting areas” and transects that did not include harvesting areas at all were defined as “non-harvesting areas”. We calculated the distance from each sampling point to the nearest human residential area using base map information developed by the Geospatial Information Authority of Japan. At each sampling point, we established quadrats (1 m × 1 m) and measured the height and coverage of each forage plant below 1.8 m above the ground. Then, we calculated the BI (plant height (cm) × coverage (cm^2^), [63]) of dwarf bamboos, broad-leaved trees, and forbs, which corresponded to plant volume. Plants that sika deer rarely forage, such as Ten’nin-sou *Comanthosphace japonica* Hooker., summer ragwort *Ligularia dentata* Gray., white hellebore *Veratrum stamineum* Maxim., hakusan shakunage *Rhododendron brachycarpum* Don., and Japanese andromeda *Pieris japonica* Don., were excluded from the calculation of BI. We classified visibility according to Takada [64] as follows: (1) very dense understory, with a visibility distance below 10 m; (2) dense understory, with a visibility distance of 10–30 m; (3) medium-density understory, with a visibility distance of 30–50 m; (4) sparse understory, with a visibility distance of 50–100 m; or (5) no understory, with a visibility distance over 100 m. We measured the visibility distances using distance meter (Nikon, COOLSHOT20, Tokyo). The mean values of the distance to the nearest human residential area, BI of each forage source, and visibility score for each transect were used to characterize the transect.

### 2.4. Statistical Analyses

The pellet groups of sika deer, forage abundances, and visibility scores analyzed in this study were published in [52,64] and we reevaluated these data. We completed the following analyses in R 4.2.2 [65] using the packages ‘spdep’ [66], ‘lme4’ [67], ‘MuMin’ [68], and ‘effects’, ‘car’ [69].

First, we checked spatial and temporal autocorrelation of the sika deer pellet group number and BI of dwarf bamboos, broad-leaved trees, and forbs, using Moran’s I test [70] and Durbin-Watson test [71], respectively. As a result, we confirmed significant positive spatial autocorrelation for all variables (sika deer fecal pellet group number: MI = 0.28, *p* < 0.001, dwarf bamboo BI: MI = 0.11, *p* < 0.01, broad-leaved tree BI: MI = 0.12, *p* < 0.01, forb BI: MI = 0.27, *p* < 0.001), but no temporal autocorrelation for all variables (sika deer fecal pellet group number: DW = 2.19, *p* = 0.45, dwarf bamboo BI: DW = 1.91, *p* = 0.74, broad-leaved tree BI: DW = 2.30, *p* = 0.24, forb BI: DW = 1.84, *p* = 0.55). Then, to examine the effect of environmental variables on spatial distribution of deer, we ran a generalized linear mixed model (GLMM1) with a Poisson error distribution and log link function. The dependent variable was the number of deer fecal pellet groups, the offset variable was log-transformed transect length, and the explanatory variables were harvesting zonation (harvesting area vs. non-harvesting area), distance to nearest human residential area, BI of dwarf bamboos, deciduous broad-leaved trees, forbs, and visibility score. To minimize the influence of outliers in explanatory variables of continuous value, after adding 0.5, they were transformed into natural logarithms [72]. Then, these explanatory variables were standardized to a mean of 0 and a standard deviation of 1 to eliminate the effect of varying measurement scales. To minimize the influence of spatial autocorrelation, the grid ID of each transect was set as a random effect. Then, we checked multicollinearity among predictor variables using variance inflation factor (VIF, collinearity considered ≥ 3). As a result, no metrics were correlated (VIF, harvesting zonation: 1.42, distance to nearest human residential area: 1.20, visibility score: 1.19, dwarf bamboo BI: 1.10, deciduous broad-leaved tree BI: 1.95, forb BI: 1.62). Akaike’s information criterion corrected for small sample size (AICc; [73]) was used to select the model that best explained the observed pattern of deer spatial distribution. Maximum likelihoods from all possible subsets of the global model containing all variables were used to obtain AICc scores and build the candidate set of explanatory models. Models with the lowest AICc values were considered the best-fit model and we obtained beta estimates of the effect of the predictor variables in the best-fit model.

To examine the effect of harvesting zonation on forage abundance for deer, we ran LMMs. The dependent variable was the BI of dwarf bamboos (LMM1), deciduous broad-leaved trees (LMM2), and forbs (LMM3). To minimize the influence of outliers in dependent variables, after adding 0.5, they were transformed into natural logarithms [72]. The explanatory variables were harvesting zonation (harvesting area vs. non-harvesting area). To minimize the influence of spatial autocorrelation, the grid ID of each transect was set as a random effect.

## 3. Results

We found fresh sika deer pellet groups in all 60 transects (100%), counting 1986 pellet groups (Appendix A). The mean ± SD pellet group number per 100 m was 6.0 ± 6.2 (range, 0.2–38.3). Model selection to identify the appropriate model structure for the number of deer pellet groups revealed that the best model included the harvesting zonation, distance to nearest human residential area, and BI of dwarf bamboos (Table 1). The best-fit model showed the following patterns: deer selected areas with non-harvesting areas (β = −0.36, SE = 0.18, z = −2.05, *p* < 0.05), far from human residential areas (β = 0.23, SE = 0.09, z = 2.56, *p* < 0.05), and higher dwarf bamboo abundance (β = 0.34, SE = 0.09, z = 3.95, *p* < 0.001, Figure 2).

The mean ± SD BI of dwarf bamboos, broad-leaved trees, and forbs were 1.9 ± 7.6, 4.5 ± 8.9, and 3.9 ± 7.6, respectively. The BI of forbs and deciduous broad-leaved trees were significantly higher in non-harvesting areas than in harvesting areas (LMM1, LRT: χ2 = 14.2, df = 1, *p* < 0.001, LMM2, LRT: χ2 = 25.9, df = 1, *p* < 0.001, Figure 3), whereas the BI of dwarf bamboos was not different between harvesting and non-harvesting areas (LMM3, LRT: χ2 = 1.1, df = 1, *p* = 0.25, Figure 3).

## 4. Discussion

In the present study, spatial distribution of sika deer was biased to non-harvesting areas, suggesting that they avoid riskier areas by following the landscape of fear concept (supporting P1). It has been shown via their within-home range habitat selection that sika deer avoid riskier open pastures during the harvesting season [28]. Similarly, the effects of a landscape of fear on habitat selection at an individual home range scale has been reported for other deer species, such as roe deer [16], red deer [17], and white-tailed deer *Odocoileus virginianus* [15]. However, the effects of human harvesting on landscape-scale spatial distribution of deer are less well understood. Our results clearly show that harvesting by humans alters the spatial distribution of sika deer. In this area, for the purpose of population control of sika deer, extremely strong harvesting pressure has been applied for more than 10 years by team-hunting maki-gari. Maki-gari have a low probability of killing all individuals in a target herd of sika deer, so surviving individuals can learn which areas are riskier. Moreover, sika deer can identify riskier harvesting areas because humans repeatedly harvest certain areas that are more accessible. Therefore, such conditions may have driven sika deer to establish a landscape of fear that influences their habitat selection at a landscape scale.

Deer avoided areas such as those near human residential areas. When deer perceived humans as a potential danger, they tended to avoid areas with high human activity [16,74]. In fact, most deer in the study area perceived humans as a potential danger, showing vigilance, flight behavior, and other antipredator behaviors in the presence of humans [55], and thus, deer may have avoided human activity in the study area. However, as will be discussed later, some deer have become habituated to the presence of humans and have started living near urban areas close to the study area in recent years [55].

In the present study, deciduous broad-leaved trees and forbs were more abundant in harvesting areas than in non-harvesting areas (supporting P2). In general, leaves of deciduous broad-leaved trees and forbs tend to be more digestible and contain more protein than graminoids [46,47], and are the preferred food items for sika deer inhabiting cool temperate zones during summer [75,76]. However, these plants tend to be less tolerant to foraging pressure, declining rapidly in areas with high densities of sika deer [19]. Therefore, in non-harvesting areas, deciduous broad-leaved trees and forbs are thought to have decreased because of severe foraging pressure by sika deer. Although these high-quality food resources were abundant in the harvesting areas, the high predation risk seems to have limited the sika deer’s use of these resources. These results suggest that there was a trade-off between predation risk avoidance and the acquisition of high-quality food resources for spatial distribution of sika deer. However, our results have a limitation, which was that we could not directly assess the effects of deer foraging pressure. In the future, it will be necessary to demonstrate the effects of foraging pressure by installing fences that remove deer foraging pressure.

Conversely, sika deer frequently used areas with abundant dwarf bamboos. Because dwarf bamboos are relatively resistant to foraging pressure by sika deer [48], this may be why they grew even in non-harvesting areas. Unlike deciduous broad-leaved trees and forbs, dwarf bamboos have a high fiber content and low digestibility [46,47]. Therefore, in general, dwarf bamboos do not tend to be a staple food for deer in the summer when the leaves of deciduous broad-leaved trees and forbs are abundant [48,75,76]. In the subalpine zone of our study area, dwarf bamboos were the main food item for sika deer throughout the year [54] and seemed to be the most important alternative food resource in non-harvesting areas where supplies of deciduous broad-leaved trees were quite limited.

Consequently, our hypothesis that human harvesting shifted the spatial distribution of sika deer to non-harvesting areas, which causes severe effects on vegetation in non-harvesting areas through foraging pressure, was supported. Moreover, sika deer seemed to avoid human activity near residential areas and favor areas with abundant dwarf bamboos as alternative food resources. In the study area, most of the harvesting areas were in the medium elevation areas, and the non-harvesting areas were in the high elevation areas from the subalpine to the alpine zones, or in the low elevations near human settlements. Therefore, our results suggest that human harvesting pressure and residences shifted the spatial distribution of sika deer from the montane forest to subalpine/alpine zones. The high elevation areas of Mt. Fuji had never been inhabited by sika deer before 2000 and are home to valuable flora and fauna [77]. Because of the invasion of sika deer in these areas, damage to ecosystems has occurred via bark stripping [21,23,24] and elimination of native Japanese serows through exploitative competition [37,54]. Furthermore, recently, sika deer live in high densities outside the study area very close to urban areas, where no harvesting has been carried out, and they are highly habituated to human presence [55]. This may also be the result of continuous strong harvesting pressure in the mid-elevation areas. A high density of sika deer near human residential areas will increase the risk of agricultural damage, traffic accidents, and zoonotic diseases [25]. The invasion of subalpine/alpine zones and the appearance near urban areas by sika deer have been reported nationwide [78], and it is possible that shifts in the spatial distribution of sika deer resulting from harvesting shown in the present study may be common to many other areas of Japan.

## 5. Conclusions

Our results suggest that human harvesting pressure and residences shifted the spatial distribution of sika deer from the montane forest to subalpine/alpine zones, which may increase damage to vulnerable (i.e., subalpine/alpine) ecosystems due to severe foraging pressure. Moreover, dwarf bamboo seemed to be important as an alternative food resource for sika deer and affects the spatial distribution of sika deer. In order to conserve subalpine/alpine ecosystems, it will be necessary to harvest sika deer in these areas.

## Figures and Tables

**Figure 1 animals-14-01924-f001:**
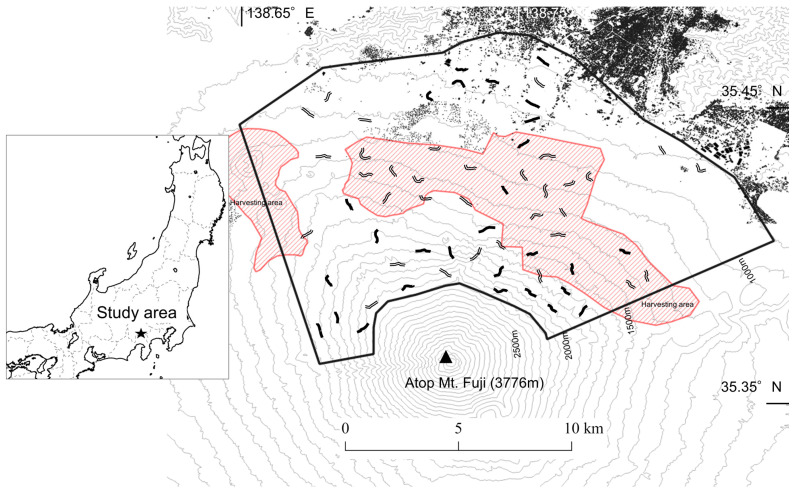
The Mount Fuji study area (black frame border) with all belt transect locations (single and double lines). Double lines indicate a sika deer *Cervus nippon* pellet group density of less than five (pellet group number/100 m), whereas single lines indicate a density of five or more (pellet group number/100 m). Areas in red with diagonal lines indicate harvesting areas, whereas white parts indicate non-harvesting areas. Black polygons indicate human residential areas.

**Figure 2 animals-14-01924-f002:**
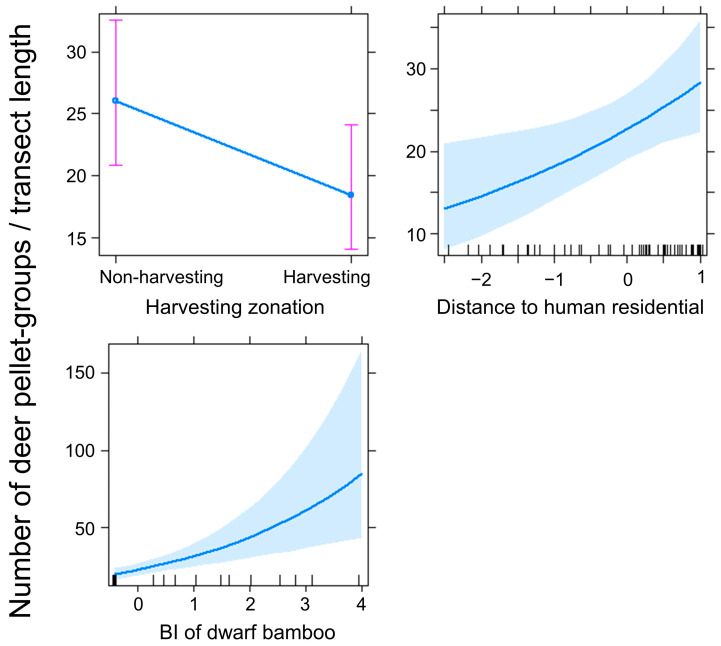
Marginal effect of explanatory variables on the best-fit model for the number of sika deer *Cervus nippon* pellet groups (± 95% CI) during summer at Mount Fuji, central Japan. Explanatory variables of nearest distance to human residential areas (m) and biomass index (BI) of dwarf bamboo were transformed into natural logarithms and standardized.

**Figure 3 animals-14-01924-f003:**
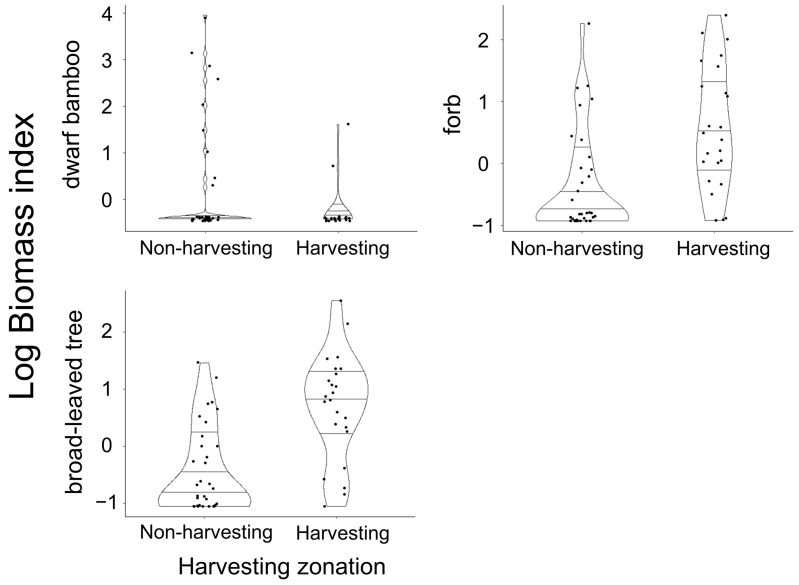
Violin plots representing the summer forage abundances of dwarf bamboos, deciduous broad-leaved trees, and forbs for sika deer *Cervus nippon* in non-harvesting areas (n = 36) and harvesting areas (n = 24) at Mount Fuji, central Japan. Points and lines inside the violin show raw data and those variations (median, range, and 25th–75th percentiles), respectively. Biomass index (BI) of deciduous broad-leaved trees were transformed into natural logarithms.

**Table 1 animals-14-01924-t001:** Model selection results (top 10 models) for the effects of various predictor variables on number of sika deer *Cervus nippon* pellet groups during summer at Mount Fuji, central Japan. The fit of each model was assessed by Akaike’s information criterion corrected for small sample sizes (AICc), ΔAICc (the difference between each model’s AICc value and the lowest AICc value), and Akaike weights (wAICc). K indicates the number of parameters in the model. Predictor variables are harvesting zonation (hv), distance to nearest human residential area (hr), visibility score (vs), biomass index of deciduous broad-leaved trees (bd), forbs (bf), and dwarf bamboos (bb). Models include various combinations of predictor variables and the null model. The best model and models with a ΔAICc of less than two are shown in bold. All predictor variables were standardized.

Model Number	Explanatory Variables	*K*	AICc	ΔAICc	wAICc
**45**	**hv + hr + bb**	**5**	**505.5**	**0.00**	**0.228**
**41**	**hr + bb**	**4**	**507.2**	**1.72**	**0.096**
**61**	**hv + hr + bf + bb**	**6**	**507.4**	**1..89**	**0.089**
47	hv + hr + vs + bb	6	507.7	2.15	0.078
57	hr + bf + bb	5	507.7	2.16	0.078
46	hv + hr + bd + bb	6	507.9	2.35	0.070
43	hr + vs + bb	5	508.7	3.17	0.047
42	hr + bd + bb	5	509.0	3.53	0.039
62	hv + hr + bd + bf + bb	7	509.4	3.87	0.033
37	hv + bb	4	509.6	4.11	0.029

## Data Availability

Data generated or analyzed during this study are included in this published article (and its Appendix A).

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
