# Peer review of "Effects of Human Harvesting, Residences, and Forage Abundance on Deer Spatial Distribution"

_animals, 2024, doi:10.3390/ani14131924_

Round 1

Reviewer 1 Report

Comments and Suggestions for Authors

Manuscript ID: animals-3042336

Title: Sika deer (Cervus nippon) spatial distribution in a landscape of fear: human harvesting drive deer to climb and descend mountains 

Authors: Hayato Takada *, Keita Nakamura

Review

In the manuscript, authors aimed to analyze the relationship between summer spatial distribution of sika deer, harvesting, and other factors, such as forage abundance, cover, and human residential areas. Investigation was done in central Japan, Mount Fuji, including areas of different elevation. Authors hypothesized that harvesting shifted the spatial distribution of sika deer from the montane forests to subalpine/alpine zones or near urban areas, which cause severe effects on vegetation in non-harvesting areas through foraging pressure. Two predictions, one about the prioritization of predation avoidance over other factors, and other about poorer food resources in non-harvesting areas, were tested.

Generally, I like this paper, and some comments about it presented below.

Title: now duplicate with keywords, just a proposal – leave only second part of the title. “Human harvesting drive deer to climb and descend mountains”.

Keywords: please edit accordingly to requirements (some text from Template remaining).

Referring: please cite by numbers, e.g. [1], [5–11}, as required by
Template

Line 56: italicize Capreolus capreolus

Lines122–123: mistake in “(P2) non-harvesting areas have poorer food resources, especially in broad-leaved trees and forbs, than non-harvesting areas due to severe foraging pressure.”?

Line 130: Figure 1

Line 181: maybe grid, not meshes?

Line 188: 1 m

Line 205: 100 m

Line 206: 6–8

Line 215: was coverage measured by cubic centimeters?

Lines 217–218: please refer to common name followed by Latin name of every species on the first use.

Figure 2: just a question – pellet groups per transect? Per 100 m? Because, see Line 275, maximum groups was 38 per 100 m.

Line 318: Bonnot, not Bonnoet, however please refer sources by numbers.

Back matter

Line 390: put table name here, as “Table S1: Raw data of the spatial distribution of sika deer in the Mount Fuji, central Japan.”.

Line 407: refer as for the role of funders, according Template.

References

Please follow format and requirements of MDPI.

Supplement

Table S1. Raw data of the spatial distribution of sika deer in the Mount Fuji, central Japan.

In general, my comments require minor revision of the text, but warrants publishing afterwards.

Comments on the Quality of English Language

I have no concerns as for the language. If necessary, small changes are facilitated by MDPI.

Author Response

Thank you for your constructive comments on our manuscript, we have made efforts to improve the manuscript. The response to the comment is below:

Title: now duplicate with keywords, just a proposal – leave only second part of the title. “Human harvesting drive deer to climb and descend mountains”.

 ↓

We have revised as your suggestion.

Changes: L2

Keywords: please edit accordingly to requirements (some text from Template remaining).

 ↓

We have revised as your suggestion.

Changes: L37-38

Referring: please cite by numbers, e.g. [1], [5–11}, as required by Template

 ↓

We have revised as your suggestion.

Changes: throughout the manuscript

Line 56: italicize Capreolus capreolus

 ↓

We have revised as your suggestion.

Changes: L51

Lines122–123: mistake in “(P2) non-harvesting areas have poorer food resources, especially in broad-leaved trees and forbs, than non-harvesting areas due to severe foraging pressure.”?

 ↓

We have revised as “(P2) non-harvesting areas have poorer food resources, especially in broad-leaved trees and forbs, than harvesting areas due to severe foraging pressure.”

Changes: L110-112

Line 130: Figure 1

 ↓

We have revised as your suggestion.

Changes: throughout Manuscript

Line 181: maybe grid, not meshes?

 ↓

We have revised as your suggestion.

Changes: throughout Manuscript

Line 188: 1 m

 ↓

We have revised as your suggestion.

Changes: L176

Line 205: 100 m

 ↓

We have revised as your suggestion.

Changes: L192

Line 206: 6–8

 ↓

We have revised as your suggestion.

Changes: L193

Line 215: was coverage measured by cubic centimeters?

 ↓

It measured by square centimeter, and we have revised as “cm2”.

Changes: L202

Lines 217–218: please refer to common name followed by Latin name of every species on the first use.

 ↓

We have added common names.

Changes: L204-206

Figure 2: just a question – pellet groups per transect? Per 100 m? Because, see Line 275, maximum groups was 38 per 100 m.

 ↓

This shows the number of pellet groups per transect length. Therefore, vertical axis of the figure has been revised.

Changes: Figure 2

Line 318: Bonnot, not Bonnoet, however please refer sources by numbers.

 ↓

 We have revised all citation and reference list according to instructions for author.

Back matter

Line 390: put table name here, as “Table S1: Raw data of the spatial distribution of sika deer in the Mount Fuji, central Japan.”.

 ↓

We have revised as your suggestion.

Changes: L372-373

Line 407: refer as for the role of funders, according Template.

 ↓

We have revised as “The funders had no role in the design of the study; in the collection, analyses, or interpretation of data; in the writing of the manuscript, or in the decision to publish the results.”.

Changes: L390-392

References

Please follow format and requirements of MDPI.

 ↓

 We have revised all citation and reference list according to instructions for author.

Changes: throughout the manuscript

Supplement

Table S1. Raw data of the spatial distribution of sika deer in the Mount Fuji, central Japan.

 ↓

We have revised as your suggestion

Changes: L372-373

Reviewer 2 Report

Comments and Suggestions for Authors

The manuscript "animals-3042336" represents a case study of a conflict of wild life and ecosystem elements. In my oppinion such kind of research is especially valuable in the modern times. The ecological equilibrium in the region was damaged and obviously not optimal managment led to the situation in the last century. The problem with large dear populations is a world wide phenomen and solving this puzzel is only seldom successful (e.g. Yellow stone). The cervids have a high reproduction rate and are consumers of big amount of plant matherial. The topic of the manuscript is of interest for a bright group of biologists, as well as managers and planners. The current situation is far from sustainable and may lead to severe damage in the sensitive area of Fuji Mountains. 

The manuscript is well prepared, based on intense field work. The methods were properly sellected and the statistics are demonstrating and supporting the habitat shift of the Sika deer. This habitat shift is related with shift in the phytocenosis of the mountain and with increased threat to the humans in the nearby settlements. The authors propose that the local predators are not able to control the population of the deer and the numbers of the harvested animals are obviously too small for reducing the population. In this case, few actions can be undertaken except restauration of predator's populations, which may be able to control the cervids. In  the present case such a restoration is hardly imaginable and the authors propose as the only possible mitigation measure the organization of deer hunt in the vicinity of the near suburbans. In my opinion in the current moment, this is the only reasonable way at least to enlarge the “fear zone” and to place it to the boarder of the settlements. In an organized societies like the Japanizes this may be managed with limited threats for the local people as not only hunting, but also different other “fear actions” (including specially developed and technological ones) may be undertaken. The drone based patrolling program may significantly contribute toward expulsion of the deer from the vicinity of the road nets and the houses.   

Minor revision proposed: 

line 135: remove the extra point

line 224: "Nikon" instead of "Nicon" 

line 306: track change

Author Response

Thank you for your constructive comments on our manuscript, we have made efforts to improve the manuscript. The response to the comment is below:

Minor revision proposed: 

line 135: remove the extra point

 ↓

We have revised as your suggestion

Changes: L123

line 224: "Nikon" instead of "Nicon" 

 ↓

We have revised as your suggestion

Changes: L212

line 306: track change

 ↓

We have revised as “Figure 3”.

Changes: L292

Reviewer 3 Report

Comments and Suggestions for Authors

This is an interesting paper on the large scale behavioural response of sika deer on Mount Fuji, where they are recently established, to the hunting zones. However, I am not convinced that the evidence is strong enough to support the conclusion. 

For delta AICs of less than two, only 2 of 3 models include 'harvesting zonation' and the plot of deer pellets (Figure 2) indicates a large overall in the CI for harvest and non-harvested zones. This implies that harvesting is not a significant driver in behaviour. Indeed the inclusion of 'hv' to the 'hr + bb' model makes a marginal improvement in the AIC. The models clearly suggest human habitation and dwarf bamboo are the main factors explaining deer pellet distribution. Thus, for example, a small correlation with altitude (and one not removed by the VIF analysis) could explain this finding, as could another factor not included in your models.

Minor comments:

Line 56 italicise Capreolus

Line 129 superscript km2

Line 178. Need to define the hunting season and hunting requirements. Are targets set - if so how? Is there a sex bias in hunting? Are there limited licenses? Does the price change each year? The readers will need to understand the hunting landscape.

Author Response

Thank you for your constructive comments on our manuscript, we have made efforts to improve the manuscript. The response to the comment is below:

For delta AICs of less than two, only 2 of 3 models include 'harvesting zonation' and the plot of deer pellets (Figure 2) indicates a large overall in the CI for harvest and non-harvested zones. This implies that harvesting is not a significant driver in behaviour. Indeed the inclusion of 'hv' to the 'hr + bb' model makes a marginal improvement in the AIC. The models clearly suggest human habitation and dwarf bamboo are the main factors explaining deer pellet distribution. Thus, for example, a small correlation with altitude (and one not removed by the VIF analysis) could explain this finding, as could another factor not included in your models.

 ↓

First of all, there was a typo in the value in Table 1 and statics of best-fit model, so we have corrected it. I apologize for this mistake. The results tend to be similar, but the inclusion of “hv” in the best model tends to significantly improve AICc (ΔAICc=1.72, wAICc=0.228). This indicates that harvesting zonation influences the spatial distribution of deer. In addition, the P-values ​​of each variable in the best-fit model are significant, and it cannot be denied that “hv” influences the spatial distribution of deer. Therefore, we did not make any major revisions to the discussion in the manuscript.

Changes: L265-267, Table1

Minor comments:

Line 56 italicise Capreolus

 ↓

We have revised as your suggestion.

Changes: L51

Line 129 superscript km2

 ↓

We have revised as your suggestion.

Changes: L117

Line 178. Need to define the hunting season and hunting requirements. Are targets set - if so how? Is there a sex bias in hunting? Are there limited licenses? Does the price change each year? The readers will need to understand the hunting landscape.

 ↓

We have added this information to Section 2.1 Study area.

Changes: L142-146

Round 2

Reviewer 3 Report

Comments and Suggestions for Authors

Thank you for addressing some of the comments on the earlier version. However, I still think that you are focussing on the least significant effect. The title refers to hunting DRIVING deer to non-hunted areas. Dwarf bamboo, and distance to human residence were both more important, so your title could be sika avoid human residences, since this finding is more significant. Also, your title implies movement of animals, whereas the results only refer to pellet counts in the main hunting season, and not in winter.

i think that you need to refocus your paper, title and discussion, toward the most significant results, and not those that seem the most appealing.
